# Quadriceps Muscle Morphology Is an Important Determinant of Maximal Isometric and Crank Torques of Cyclists

**DOI:** 10.3390/sports11020022

**Published:** 2023-01-18

**Authors:** Fábio Juner Lanferdini, Fernando Diefenthaeler, Andressa Germann Ávila, Antônio Renato Pereira Moro, Stephan van der Zwaard, Marco Aurélio Vaz

**Affiliations:** 1Biomechanics Laboratory, Center of Physical Education and Sports, Universidade Federal de Santa Maria, Santa Maria 97105-900, RS, Brazil; 2Biomechanics Laboratory, Center of Physical Education and Sports, Universidade Federal de Santa Catarina, Florianópolis 88040-900, SC, Brazil; 3Department of Production Engineering, Center of Technology Center, Universidade Federal de Santa Maria, Santa Maria 97105-900, RS, Brazil; 4Laboratory for Myology, Department of Human Movement Sciences, Faculty of Behavioural and Movement Sciences, Vrije Universiteit Amsterdam, 1081 BT Amsterdam, The Netherlands; 5Exercise Research Laboratory, School of Physical Education, Physiotherapy and Dance, Universidade Federal do Rio Grande do Sul, Porto Alegre 90690-200, RS, Brazil

**Keywords:** cyclists, performance predictors, maximal knee extensor torque, crank torque, quadriceps muscle properties

## Abstract

The aim of this study was to determine if quadriceps morphology [muscle volume (MV); cross-sectional area (CSA)], vastus lateralis (VL) muscle architecture, and muscle quality [echo intensity (ECHO)] can explain differences in knee extensor maximal voluntary isometric contraction (MVIC), crank torque (CT) and time-to-exhaustion (TTE) in trained cyclists. Twenty male competitive cyclists performed a maximal incremental ramp to determine their maximal power output (PO_MAX_). Muscle morphology (MV; CSA), muscle architecture of VL and muscle quality (ECHO) of both quadriceps muscles were assessed. Subsequently, cyclists performed three MVICs of both knee extensor muscles and finally performed a TTE test at PO_MAX_ with CT measurement during TTE. Stepwise multiple regression results revealed right quadriceps MV determined right MVIC (31%) and CT (33%). Left MV determined CT (24%); and left VL fascicle length (VL-FL) determined MVIC (64%). However, quadriceps morphological variables do not explain differences in TTE. No significant differences were observed between left and right quadriceps muscle morphology (*p* > 0.05). The findings emphasize that quadriceps MV is an important determinant of knee extensor MVIC and CT but does not explain differences in TTE at PO_MAX_. Furthermore, quadriceps morphological variables were similar between the left and right quadriceps in competitive cyclists.

## 1. Introduction

Endurance cycling performance is determined by physiological factors, such as maximal oxygen uptake (VO_2MAX_), physiological transition thresholds, and metabolic efficiency (e.g., cycling economy, gross efficiency) [1,2,3]. VO_2MAX_ is probably the most tested determinant for cycling performance [4,5]. Moreover, neuromuscular parameters have also been used to determine cycling time-to-exhaustion (TTE) and performance [6]. Miller and Manfredi [7] showed that physiological (i.e., anaerobic threshold) and anthropometric (i.e., thigh + calf/arm + chest) parameters are important performance determinants during a cycling time-trial (TT). Cycling endurance performance during TT is also explained by oxygen consumption (VO_2_), muscular hemoglobin concentration, and skeletal muscle oxygenation [8,9]. In addition, Lanferdini et al. [10] showed that the mechanical resultant pedaling force is a determinant of cycling submaximal performance as measured by the power output (PO). Additionally, other studies have found moderate or strong correlations between physiological [11,12,13], or neuromuscular [14] variables with cycling endurance TT performance.

Furthermore, a recent investigation showed that quadriceps muscle volume (MV) and vastus lateralis (VL) pennation angle (PA) determined 76% and 11% of the variance in peak power output (PPO) during sprint tests, respectively [15]. Therefore, in elite cyclists, VL-MV, in combination with the percentage of type-II muscle fibers, explained 65% of variance in PPO during a Wingate test [8]. Focusing specifically on knee extensor function, MV of the quadriceps femoris seems to be the best predictor (explaining 60%) of isometric knee extensor torque [16], and isokinetic PO during knee extension (explaining ~80%) in healthy subjects [17]. It has been suggested that MV is a determinant of maximal voluntary isometric contraction (MVIC) or crank torque (CT) in cyclists, but this remains to be determined.

In addition, previous studies found a negative correlation between the echo intensity (ECHO; lower values mean better muscle quality) of rectus femoris (RF) and knee extensors’ torque [18]. Similar findings of a negative correlation between ECHO from gastrocnemius lateralis and soleus muscles and a positive correlation between MV of the triceps surae muscles [19] and the triceps surae’s torque [19] have been reported in the literature. These results demonstrate that the lower the ECHO is (better muscle quality), the greater the torque production capacity of the assessed muscles [18,19]. However, the sample of both studies was composed of healthy subjects (men and women), non-cyclists or the elderly [16,17,18,19]. Furthermore, Song et al. [20] showed a moderate non-significant relationship between quadriceps ECHO (RF and vastus intermedius—VI) with maximum knee extensor strength. These results demonstrate a contradiction between ECHO and the ability to produce muscle strength. Moreover, no study was found relating ECHO to force production capacity during crank cycle.

Although some studies have verified a relationship between muscle morphology (e.g., physiological cross-sectional area—PCSA), muscle architecture (fascicle length—FL) and skeletal muscle respiration with cycling aerobic performance [2,8,9,21], to date, no evidence has been found using different variables of quadriceps femoris muscle morphology [i.e., MV, cross-sectional area (CSA)], muscle quality (i.e., ECHO), and VL muscle architecture to determine MVIC and CT during TTE in cyclists, nor have we found studies assessing a possible relationship between quadriceps muscle morphology, muscle quality and muscle architecture and TTE performance in cyclists. Therefore, the objective of this study was to determine if knee extensor MV, CSA, ECHO and VL muscle architecture were able to determine MVIC, CT and TTE performance in cyclists. If indeed cycling performance is somehow determined by quadriceps morphology, coaches and cyclists may decide whether to allocate time for training-specific strength exercises aimed at quadriceps muscle hypertrophy during their regular endurance cycling training.

## 2. Materials and Methods

### 2.1. Experimental Approach

We carried out a cross-sectional study to understand if quadriceps morphology and quality, and VL muscle architecture, are determinants of maximal isometric knee extensor torque and of crank torque during TTE performance. Each cyclist visited the laboratory on two occasions (Figure 1). During the first visit, anthropometric data were assessed, and cyclists performed a maximal incremental test and familiarization to maximal TTE. After a week, athletes returned for the second visit. Muscle morphology (estimated MV; CSA), muscle quality (ECHO) and VL muscle architecture of both the left and right quadriceps’ muscles were assessed. After that, cyclists performed three knee extensor MVICs with both lower limbs. Finally, cyclists performed a maximal TTE at maximal power output (PO_MAX_) with measured CT. This study was conducted according to the Declaration of Helsinki, and all procedures were approved by the local Institutional Research Ethics Committee (project number 708.362). All cyclists were informed of the benefits and risks of the investigation prior to signing an institutionally-approved informed consent document to participate in the study. Before each visit, subjects were instructed to avoid strenuous exercise and alcohol consumption within the last 48 h and to consume no caffeine or food during the final 3 h before each test. The athletes participating in the present study had 5.8 ± 6.6 years of regular training/competition and no history of lower limb muscle-skeletal injuries. Exclusion criteria included chronic disease, smoking, metabolic disorders, use of steroids in the last six months, chronic disease, physical disabilities, smoking, and use of antibiotic drugs in the previous week.

### 2.2. Participants

Twenty endurance-trained male cyclists participated in the study, having the following physical and physiological characteristics: Age 29.2 ± 6.6 years; body mass 77.1 ± 10.5 kg; height 179 ± 8 cm; PO_MAX_ 377.6 ± 34.5 W; VO_2MAX_ 57.0 ± 7.7 mL∙kg∙min^−1^; training volume 4.6 days and ~264 km/week; classified as performance level 3 (trained), according to De Pauw et al. [22]. Cyclists competed at the regional and national levels.

### 2.3. Procedures

During the first session, anthropometric data were measured according to the International Society for the Advancement of Kinanthropometry [23]. After that, cyclists performed a warm-up with 150 W of workload for 10 min. Cyclists were tested using a standard road cycling bicycle (Giant TCR Advanced, Taichung, Taiwan) with handlebars configuration and saddle position set to their anthropometrical characteristics. The bicycle was mounted on a stationary cycling trainer (CompuTrainer, ProLab 3D, Racermate Inc., Seattle, WA, USA) to determine PO_MAX_ (in Watts). Before testing, tire pressure was calibrated according to manufacturer instructions (~100 psi). Laboratory temperature (26–28 °C) and humidity (~50%) were controlled during all testing to minimize temperature effects on bicycle tire pressure and PO measurements [24]. Cyclists performed an incremental ramp test with 25 W increments every minute (~0.42 W/s) until exhaustion, using a custom-made script in cycling trainer software (CompuTrainer, CS 1.6, Racermate Inc, Seattle, WA, USA). Cadence was maintained close to 95 ± 5-rpm for all cyclists, using visual feedback from the cycling trainer control unit. Exhaustion was defined by the following criteria: voluntary exhaustion or cadence dropping below 70 rpm. VO_2_ was measured by an open-circuit indirect gas exchange system (CPX/D, Medical Graphics Corp., St. Louis, MO, USA) and VO_2MAX_ was defined as the greatest value obtained in the last stage of the incremental test, along with PO_MAX_. After incremental tests, cyclists pedaled for ~30 min at 50 W for recovery purposes and, finally, cyclists performed a familiarization with the TTE at PO_MAX_ and a 95 ± 5-rpm of cadence.

In the second session, quadriceps muscle morphology, muscle architecture and muscle quality were measured by the same investigator with extensive experience with ultrasonography acquisition of muscles (~10 years). Quadriceps ultrasonography images were acquired using a B-mode Aloka ultrasound system (SSD 4000; ALOKA, Tokyo, Japan) with a 60-mm linear array transducer and 7.5 MHz. The ultrasonography probe was coated with a water-soluble gel to provide acoustic contact and was positioned on the skin without depressing the dermal surface. All ultrasonography images were acquired at rest with the lower limbs fully extended, after subjects rested for 10 min in a supine position on a stretcher. Three transversal ultrasound images were obtained for each muscle [VL, VI and RF, as well as the quadriceps muscle thickness (MT)] from both quadriceps’ muscles (right and left). The probe was placed transversally (50% of the distance between the greater trochanter and the lateral femur condyle) using fixed settings on the ultrasound equipment (frequency: 7.5 MHz; depth: 8 cm; General Gain: 40 dB; Time Gain Compensation—TGC in neutral position and focal zone 1.0 cm). After that, three longitudinal ultrasound images were obtained of VL (right and left) from each cyclist. After that, three longitudinal ultrasound images were obtained of VL (right and left) from each cyclist, with the probe placed longitudinally to the muscle at 50% of the distance between the greater trochanter and the lateral femur condyle. Femur length was measured using a metric fiberglass tape (Sanny, São Bernardo do Campo, Brazil, with 1 mm precision) from the distance between the femur’s greater trochanter and the articular cleft between the femur and tibia condyles [25].

Quadriceps MV was estimated from the MT measurement between RF’s superficial aponeurosis and VI’s deep aponeurosis using ImageJ 1.42q software (National Institute of Health, Bethesda, MD, USA). Quadriceps MV was estimated using the equation proposed by Miyatani et al. [25], where: Quadriceps MV = [(Quadriceps MT (RF + VI) ∙ 320.6) + (femur length ∙ 110.9) − 4437.9].

All ultrasound images were analysed by the same investigator with extensive experience using ultrasonography analysis with the ImageJ 1.42q software (National Institute of Health, Bethesda, MD, USA). A maximum region of interest captured by the ultrasonography probe (60-mm) was determined in each muscle and used to determine the quadriceps CSA and ECHO [26,27]. CSA measurements may have been underestimated due to the CSA’s size of the assessed muscles, which, in some cases, exceeded the image’s area captured by the ultrasound probe (60-mm). Mean grayscale of ECHO value of each muscle was determined using a standard grayscale histogram function and expressed as a value between 0 (black) and 255 (white) in the same software. The mean of three ultrasound images was used to quantify quadriceps CSA (RF + VI + VL) and ECHO [(RF + VI + VL)/3]; Figure 2. In addition, the VL best fascicle (i.e., the fascicle that was fully visible from its insertion on the deep aponeurosis to the superficial aponeurosis, or to the ultrasound probe field-of-view end) in each ultrasonography image was used for muscle architecture analysis. FL was considered the length of the fascicular path between superficial and deep aponeuroses. When the ends of the fascicles were outside the ultrasound image, FL was estimated from extrapolation, as recommended in a previous study [28]. PA was calculated as the angle between the muscle fascicle and the deep aponeurosis. MT was considered a straight line between the deep and superficial aponeurosis along each ultrasonography image (Figure 2). Mean values were obtained from three ultrasound images for each muscle to determine FL, PA and MT of VL. The error in estimating the entire FL using the linear model ranged from 2–7% [28] to 13% [29].

After that, athletes were asked to sit on a chair of an isokinetic dynamometer (Biodex System 3 Pro, 2000 Hz, Biodex Medical Systems, Shirley, NY, USA) to perform MVIC of the knee extensor muscles, which was evaluated at 70° of knee flexion (0° = full knee extension). After fixation of the subject on the dynamometer chair, a verbal encouragement was given by researchers in each MVIC so that cyclists performed maximal torque in all contractions. All participants performed three 5-sec MVICs, with a 2-min rest interval between contractions. MVICs were measured from both lower limbs, and the highest or peak MVIC from each limb was used for further analysis.

Finally, cyclists performed a maximal TTE at PO_MAX_ with the road bike coupled to a stationary cycling trainer (CompuTrainer, ProLab 3D, Racermate Inc., Seattle, WA, USA). During TTE, instrumented cranks (MEP, Studio AIP SRL, 2 Hz, Oggiona con Santo Stefano, Italy) were used to measure CT in both lower limbs, with the software MEP Studio AIP (MEP Manager 1.6—Studio AIP SRL, Oggiona con Santo Stefano, Italy). The mean CT of each crank (right and left) during the TTE was used for further analysis.

### 2.4. Statistical Analysis

Data normality, homoscedasticity and sphericity were assessed by the Shapiro-Wilk, Levene and Mauchly tests, respectively. Stepwise multiple linear regressions were used to estimate the relative contributions of independent morphological (CSA, MV), architectural (VL’s MT, PA and FL) and muscle quality (ECHO) variables of both quadriceps on the dependent variables of performance (knee extensors MVIC, CT and TTE). Our collinearity diagnostic exploration resulted in variance inflation factors (VIF) of <2.0 and tolerance of 0.10–0.70, which indicate acceptable levels of multicollinearity of the independent variables [30]. In addition, post-hoc power of the multiple linear regression was calculated according to Cohen et al. [31]. Effect size (ES) of multiple linear regression was calculated and classified as small (>0.02); moderate (>0.13); and large (>0.26) a priori, using G*Power 3.1.9.7 (FrauzFaurUniversität, Kiel, Germany), as described by Faul et al. [32]. The correlation matrix showed the magnitude of Pearson’s product-moment correlation coefficient between dependent variables (knee extensors MVIC, CT and TTE), and independent muscle variables [CSA, MV, ECHO and VL muscle architecture (MT, PA and FL)]. The correlation was classified as small (R = 0.0–0.1); moderate (R = 0.1–0.3); large (R = 0.3–0.5); very large (R = 0.5–0.7); and extremely large (R = 0.9–1.0), according to thresholds recommended by Hopkins et al. [33] using R-Studio (R version 4.1.0, R Core Team, 2021). Dependent Sample *t*-test was used to compare sides. Simple linear regressions were performed to verify the relationship between sides for all investigated variables. All statistical analysis was performed with SPSS 22.0 for Windows (IBM SPSS Inc, Chicago, IL, USA), with a significance level of α = 0.05. All dataset is in Supplementary Material (Appendix A).

## 3. Results

The cyclists presented the following TTE performance at PO_MAX_: 150.8 ± 38.2 s. Initially, stepwise multiple linear regression analyses of multicollinearity excluded from the regression models all variables that presented below 2 VIF and tolerance outside 0.10–0.70. The stepwise multiple linear regression model demonstrated that the right quadriceps MV is an important determinant of right MVIC (31.0%) and CT (32.6%) of cyclists (Table 1). Moreover, left quadriceps MV and left vastus lateralis fascicle length (VL-FL) were determinants of left MVIC (64.2%), and left quadriceps MV was an important determinant of left CT (23.5%) of cyclists (Table 1). However, no variables were able to determine the TTE at PO_MAX_ of cyclists.

Right (R = 0.56) and left (R = 0.59) quadriceps MV presented very large positive correlations with right and left MVIC, respectively (*p* < 0.05). Furthermore, right (R = 0.57) and left (R = 0.49) quadriceps MV presented very large and large positive correlations with right and left CT, respectively (*p* < 0.05). In addition, left quadriceps ECHO (R = 0.47) presented large positive correlation with left CT, and left VL-FL (R = 0.63) presented very large positive correlation with left MVIC (*p* < 0.05) (Figure 3).

Table 2 shows comparisons between left and right lower limbs for dependent (MVIC and CT) and independent [Quadriceps MV, CSA, ECHO and muscle architecture of VL (MT, PA and FL)] variables. No significant between-sides differences were found for CSA, ECHO, vastus lateralis pennation angle (VL-PA), VL-FL and MVIC (*p* > 0.05). However, the right lower limb presented greater quadriceps MV and vastus lateralis muscle thickness (VL-MT), and smaller CT, compared to the left lower limb (*p* < 0.05).

Simple linear regressions showed significant relationships between left and right limbs for all variables [MVIC (R^2^ = 0.76; *p* < 0.001); CT (R^2^ = 0.79; *p* < 0.001); quadriceps MV (R^2^ = 0.94; *p* < 0.001); quadriceps CSA (R^2^ = 0.59; *p* < 0.001); VL-MT (R^2^ = 0.76; *p* < 0.001); VL-PA (R^2^ = 0.27; *p* = 0.012); VL-FL (R^2^ = 0.27; *p* = 0.012); and quadriceps ECHO (R^2^ = 0.88; *p* < 0.001)]; Figure 4.

## 4. Discussion

The purpose of this study was to identify which parameters of quadriceps muscle morphology size (MV and CSA), quadriceps muscle quality (ECHO) and VL muscle architecture (FL, PA and MT) from both lower limbs are determinants of knee extensor MVIC, CT (both sides) and TTE performance in trained cyclists. Our outcomes showed large ES (≥0.31) for the stepwise multiple linear regression model and large observed power (≥0.81), demonstrating adequate sample size and confirming the cause-effect of the all-regression analyses performed. According to our results, 31% of right knee extensor MVIC was determined by right quadriceps MV, whereas 33% of right CT was determined by right quadriceps MV. In addition, 64% of left knee extensor MVIC was determined by left quadriceps MV and left VL-FL, whereas 24% of left CT was determined by left quadriceps MV. However, the evaluated quadriceps morphological parameters did not explain differences in the TTE performance at PO_MAX_ in competitive cyclists. Additionally, our results showed strong positive correlations between right and left quadriceps MV with knee extensor MVIC and CT of both lower limbs. Furthermore, left quadriceps ECHO presented positive correlations with left CT, and left VL-FL presented positive correlations with left knee extensor MVIC. Results showed no between-sides differences for CSA, ECHO, VL-PA, VL-FL and MVIC. However, the right limb presents greater quadriceps MV (2.8%) and VL-MT (5.5%) and smaller CT (3.8%) compared to the left limb of cyclists. These results suggest no asymmetries between lower limbs [34]. Carpes et al. [34], in their review, showed that cyclists exhibit higher asymmetry indexes for CT or PO during moderate to low intensity exercise, but intensities eliciting maximal effort (e.g., PO_MAX_) were suggested to be symmetric between lower limbs. However, there is a lack of investigations about muscle morphology, muscle architecture, muscle quality, and muscle activation that allow us to determine the origins of possible asymmetries.

Our results agree with previous studies, where MV (76%) predicted the PPO during sprints tests [15] and was the best determinant (60%) of knee extensor MVIC torque [16]. Furthermore, in a group of elite cyclists, VL MV, in combination with the percentage of type-II muscle fibers, explained 65% of the variance in PPO during a Wingate test [8]. Our results showed that the quadriceps MV of endurance cyclists determine ~30% of knee extensor MVIC. However, these MV values are well below those of sprinter track cyclists (76%) to determine PPO during sprint tests [15]. One of the explanations may be related to the fact that long-distance cyclists depend too highly on physiological (e.g., energetic) conditions to improve their performance when compared to sprinters [1,2,3,4,5,6,7,8,9]. In addition, our results demonstrate that the left VL-FL helped explain differences in the left knee extensor MVIC. These results suggest that cycling training may have generated musculoskeletal adaptations on VL-FL and consequently increased the athlete’s capacity to produce maximum torque at the knee extensors’ optimal angle (~70° of knee flexion) [35,36]. However, the same result was not found for the right side, which may be due to variability in the ultrasonography measurements of the VL muscle (Figure 4). Moreover, Kordi et al. [15] showed that the quadriceps VL-PA determined 11% of PPO during sprint tests, which disagrees with our findings, most likely due to the type of test performed. Furthermore, our results demonstrate that no quadriceps muscle morphology and quadriceps muscle quality and/or VL muscle architecture variables was able to explain differences in the TTE performance at PO_MAX_ in endurance cyclists, probably related to the fact that all athletes were pedaling at maximum aerobic workload. Perhaps if the TTE tests were performed at a submaximal workload (e.g., 80% of PO_MAX_) the results could be different, and better conditioned cyclists would have a higher performance, which is not necessarily the case when the workload is relative to 100% (i.e., maximal). Therefore, we suggest that future studies also test the quadriceps morphology, muscle quality and muscle architecture as performance determinants in submaximal aerobic tests (Workload Absolute or Relative).

However, despite the strong relationships between MVIC and CT with MV and VL-FL, muscle morphology may also interact with other variables such as neuromuscular [6], cardiac, skeletal, and anthropometric [7] parameters. van der Zwaard et al. [8] evaluated the effect of several independent variables (oxygen consumption, blood sampling, knee-extensor maximal force, muscle oxygenation, muscle morphology, and muscle fiber histochemistry of VL) in the cyclists’ performance, and suggested that VL-FL and capillarization are important targets for training to optimize sprint and endurance performance simultaneously. Furthermore, muscle morphology of multiple muscles involved on crank cycling (recently investigated in a simulation model) showed that better alignment of the peak power-pedaling rate curve of the vasti muscles may improve cycling PPO [37]. In the present study, other muscle groups that almost certainly contribute to CT, such as gluteus maximus and plantar flexors, were not assessed [38]. Assessment of other major muscle groups would have given a more complete understanding of muscular morphological, quality, and architectural determinants of CT during cycling. Nevertheless, our findings for the predominant influence of quadriceps femoris MV on MVIC and CT reinforce the importance of muscle size (~30%) for neuromuscular force/power production. Our results also suggest that cyclists and coaches should be especially attentive to strength training and nutrition strategies to enhance MV. Plyometric and resistance training are well-known strategies to stimulate hypertrophy and increase MV [39,40], and could be used to improve the cyclists’ performance.

Furthermore, our results demonstrated a large positive correlation between both sides of quadriceps ECHO with both sides of knee extensors MVIC and between left quadriceps ECHO and left CT (despite not entering the stepwise multiple linear regression model), without correlations between right quadriceps ECHO and right CT, and both sides of quadriceps ECHO with both sides of MVIC of knee extensors. These results contradict previous studies, which found negative or no correlations [18,19,20] between these outcomes. In addition, the fact that our study investigated endurance cyclists may have contributed to the differences between studies. It is possible that different long-term physical demands (in our case, endurance) may have determined specific changes in strength and muscle quality. More specifically, slow-twitch fibered muscles may depend more on the connective tissue (e.g., epimysium, perimysium, endomysium, fascia and tendon) to transmit force during long cycling periods, thereby increasing the ECHO in the muscle belly. Athletes with a higher force production capacity (i.e., higher in parallel sarcomeres or myofibrils within their muscle fibers) may also need a higher connective tissue content within the muscle, thereby explaining the positive and large correlation between ECHO and MVIC in both sides. However, no previous studies were found that compared muscle quality between different sports or related ECHO with the capacity to produce maximum or specific force in athletes. We also evaluated mean quadriceps ECHO [(RF + VI + VL)/3], unlike previous studies that evaluated one or two knee extensors. Our results suggest that the use of ECHO for inferring muscle quality should be revisited [20], especially with long-distance or endurance athletes. However, our study has a small and possibly heterogeneous sample (e.g., heterogeneous TTE performance), which is a limitation of our study and, therefore, our results should be looked at with care.

Additionally, the ability of pennate muscles (i.e., VL) to undergo muscle architectural dynamic changes to amplify fiber contraction speed (i.e., amplify quadriceps fiber contraction speed during force-velocity changes that occur during the crank cycle) present significant interactions with muscle architectural gearing during contractions [41]. Pennate muscles (e.g., VL) change thickness and width during shortening [42,43]. Increasing thickness increases the amount of fiber rotation during shortening, and consequently increases muscle architectural gear ratio, whereas decreasing thickness decreases fiber rotation and muscle architectural gear ratio [43,44]. Variable gearing occurs at the level of the tendon and may be due to changes in muscle coordination across a different range of locomotor tasks [45]. Therefore, greater PA will result in shorter fascicles (due to the arrangement of muscle origins and insertions) and, consequently, the muscle will have a slower maximum shortening velocity [46]. Furthermore, short fascicles, shortening with the same absolute velocity as long fascicles, imply greater relative shortening velocities compared to those of long fascicles, and therefore a subsequently greater reduction in the force potential according to the force-velocity relationship [47]. Indeed, it is well known that muscles with greater PA may operate with higher belly gearing, which is associated with higher strength potentials [47]. Moreover, dynamic interplay between fiber force and connective tissue behavior (e.g., tendon) determines the muscle gear [41,43]. However, we did not find studies that have investigated the relationship between architectural gear ratio and torque or PO in cycling. Therefore, further studies should investigate the relationship between muscle architectural gear ratio and torque or PO during cycling.

One limitation of the present study was not evaluating the vastus medialis muscle morphology, especially for calculating the total CSA of the quadriceps muscle. Furthermore, the size of the probe (60-mm) did not allow us to capture the total CSA from all the evaluated muscles, thereby limiting our quadriceps CSA results. Discrepancies in TTE performance at PO_MAX_ may also have affected the results of the present study. In addition, we did not evaluate other cycling tests (i.e., incremental or TT) to elucidate which are the possible quadriceps’ architectural, morphological and/or quality indicators that determine the torque applied to the crank cycle. Another limitation is that we used only transverse ultrasound images to estimate the quadriceps MV, whereas the “gold standard” would be the evaluation by magnetic resonance imaging (MRI), or using 3D ultrasound, having the benefits like those of MRI [48,49]. Nonetheless, the 2D ultrasound evaluation that was applied has been validated by previous studies [26].

### Practical Application

Understanding which variables can be predictors of maximal quadriceps isometric torque and crank torque during cycling, may help coaches and athletes to be aware of some of the physical characteristics needed for the best performance (e.g., applied resistance training for improving quadriceps MV, and, consequently, the cyclists’ performance). However, they should also be aware of the simultaneous nature of cycling training and resistance exercises, which can mitigate muscle hypertrophic responses [50]. In this study, MV and VL-FL were shown to be important morphological determinants of knee extensor MVIC and CT, but not in TTE performance at PO_MAX_ of endurance cyclists. The next decisive step will be to investigate the prediction of different morphological variables on cycling endurance performance during a race (i.e., TT).

## 5. Conclusions

Quadriceps MV was an important determinant of knee extensor MVIC and CT in both lower limbs of cyclists. Left VL-FL also determines the left knee extensor MVIC. However, no quadriceps morphological variables explained differences in the TTE performance at PO_MAX_ in cyclists. Our results showed consistent and similar muscle morphology between the right and left limbs and revealed strong positive correlations.

## Figures and Tables

**Figure 1 sports-11-00022-f001:**
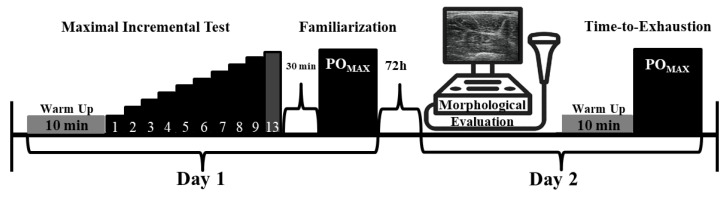
Experimental design. Maximal power output (PO_MAX_).

**Figure 2 sports-11-00022-f002:**
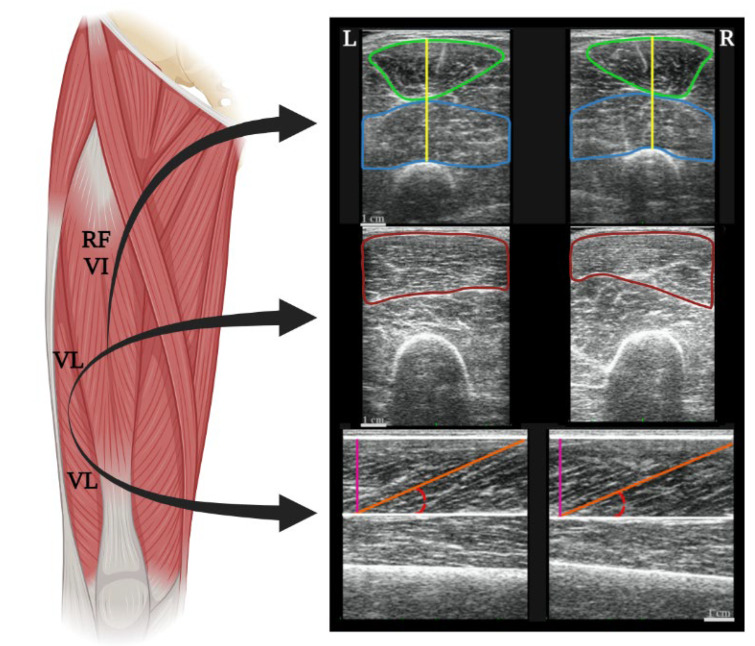
Illustration of left (L) and right (R) quadriceps muscle architecture. Both upper images represent the cross-sectional area (CSA) of the rectus femoris muscles—RF (green line), vastus intermedius—VI (blue line) and muscle thickness of the RF and VI muscles (yellow line) for later calculation of muscle volume (MV). Both middle images represent the CSA of the vastus lateralis—VL (brown line). Both inferior images represent the VL muscular architecture. The superior and inferior aponeuroses are represented by the white lines, while the muscular fascicle length (FL) is represented by the orange line, the pennation angle (PA) by the red line, and muscle thickness (MT) by the pink line. All analyses were performed using ImageJ 1.42q software (National Institute of Health, Bethesda, MD, USA).

**Figure 3 sports-11-00022-f003:**
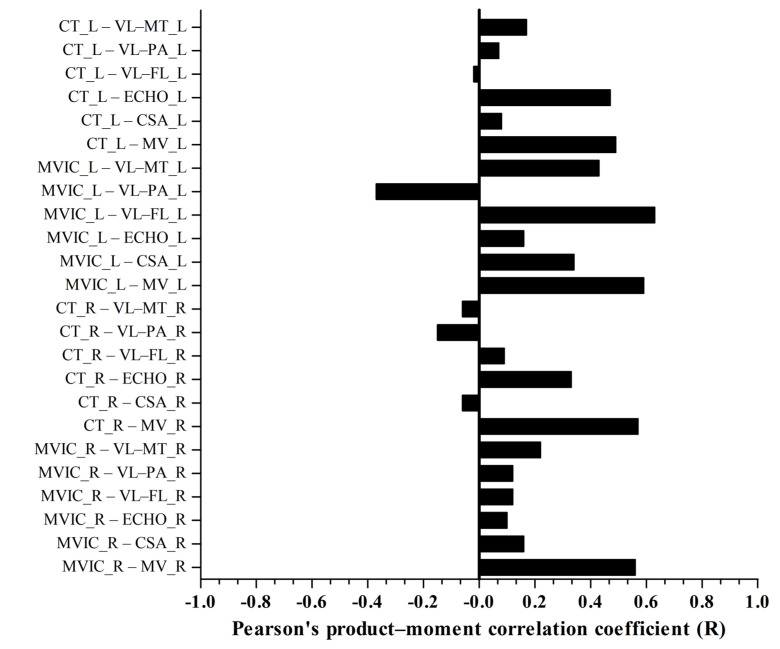
Correlation matrix showing the relationship between all dependent and independent variables. Right knee extensor maximal voluntary isometric contraction (MVIC_R); left knee extensor maximal voluntary isometric contraction (MVIC_L); right crank torque (CT_R); left crank torque (CT_L); right quadriceps muscle volume (MV_R); left quadriceps muscle volume (MV_L); right quadriceps cross-sectional area (CSA_R); left quadriceps cross-sectional area (CSA_L); right vastus lateralis muscle thickness (VL-MT_R); left vastus lateralis muscle thickness (VL-MT_L); right vastus lateralis pennation angle (VL-PA_R); left vastus lateralis pennation angle (VL-PA_L); right vastus lateralis fascicle length (VL-FL_R); left vastus lateralis fascicle length (VL-FL_L); right quadriceps echo intensity (ECHO_R); and left quadriceps echo intensity (ECHO_L). Independent variables right (R = 0.56) and left (R = 0.59) quadriceps MV presented very large positive correlation with right and left MVIC, respectively (*p* < 0.05). Right (R = 0.57) and left (R = 0.49) quadriceps MV presented very large and large positive correlations with right and left CT, respectively (*p* < 0.05). Left quadriceps ECHO (R = 0.47) presented large positive correlation with left CT, and left VL-FL (R = 0.63) presented very large positive correlation with left MVIC (*p* < 0.05).

**Figure 4 sports-11-00022-f004:**
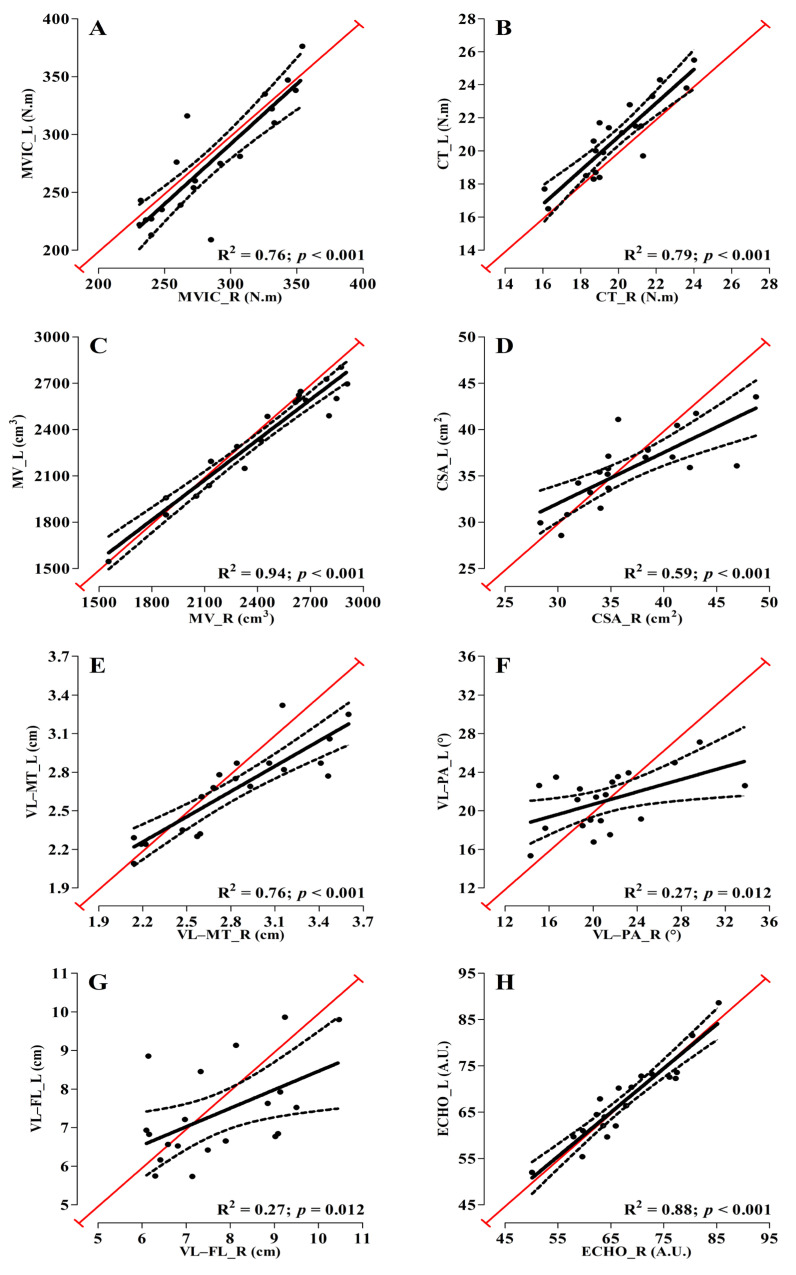
Between-sides simple linear regressions (black line) for the following variables: (**A**) knee extensors maximal voluntary isometric contraction (MVIC); (**B**) crank torque (CT); (**C**) quadriceps muscle volume (MV); (**D**) quadriceps cross-sectional area (CSA); (**E**) vastus lateralis muscle thickness (VL-MT); (**F**) vastus lateralis pennation angle (VL-PA); (**G**) vastus lateralis fascicle length (VL-FL); and (**H**) quadriceps echo intensity (ECHO). Red line represents perfect linear relationship between sides for each variable.

**Table 1 sports-11-00022-t001:** Determinants of knee extensors maximal voluntary isometric contraction (MVIC) and crank torque (CT) of cyclists.

Dependent Variable	R^2^	Indicator	Standardized Coefficients (β)	*p*-Value	Effect Size	Observed Power
MVIC right	0.310	MV right	0.557	0.011	0.45	0.81
MVIC left	0.642	MV left VL-FL left	0.502 0.553	0.003 0.002	1.79	0.84
CT right	0.326	MV right	0.571	0.009	0.48	0.82
CT left	0.235	MV left	0.485	0.030	0.31	0.81

Quadriceps muscle volume (MV); left vastus lateralis fascicle length (VL-FL left).

**Table 2 sports-11-00022-t002:** Between-sides comparison for quadriceps muscle morphology (Dependent Sample *t*-test).

Variables	Left	Right	Differences (%)	*t*-test	*p*-Value
CSA (cm^2^)	35.8 ± 4.0	36.9 ± 5.5	2.9 ± 9.7	1.341	0.196
ECHO (A.U.)	67.5 ± 8.7	67.7 ± 8.6	0.4 ± 4.7	0.304	0.765
MV (cm^3^)	2357 ± 345	2427 ± 385	2.8 ± 4.2	3.095	0.006
VL-MT (cm)	2.7 ± 0.3	2.8 ± 0.5	5.5 ± 8.3	2.942	0.008
VL-PA (°)	21.0 ± 3.0	21.0 ± 4.8	1.2 ± 19.2	0.161	0.874
VL-FL (cm)	7.4 ± 1.3	7.7 ± 1.3	6.2 ± 17.1	1.267	0.220
MVIC (N·m)	275.2 ± 50.5	284.0 ± 42.4	4.2 ± 10.1	1.594	0.127
CT (N·m)	20.8 ± 2.4	19.9 ± 2.1	−3.8 ± 5.2	−3.472	0.003

CSA: quadriceps muscle cross-sectional area; ECHO: quadriceps muscle echo intensity; MV: quadriceps muscle volume; VL-MT: vastus lateralis muscle thickness; VL-PA: vastus lateralis pennation angle; VL-FL: vastus lateralis fascicle length; MVIC: knee extensors maximal voluntary isometric contraction; CT: crank torque; TTE: time-to-exhaustion. Significant between-sides differences (*p* < 0.05).

## Data Availability

The data that support the outcomes of the present study are available online at FigShare since 14/11/2022, see the link: https://doi.org/10.6084/m9.figshare.14449638.

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
