# Peer review of "Quadriceps Muscle Morphology Is an Important Determinant of Maximal Isometric and Crank Torques of Cyclists"

_sports, 2023, doi:10.3390/sports11020022_

Round 1

Reviewer 1 Report

Dear authors,

Thank you for presenting this study on the contribution of quadriceps muscle morphological, architectural and structural parameters as determinants of knee extensor maximal isometric torque, crank torque and time to exertion performance in trained cyclists. 

Overall, the message of the study is relatively clear, but there are still quite some spelling mistakes and several paragraphs lack structure. Some of the errors are listed below, but I recommend the writers thoroughly read through the article again after resubmitting (or let it be checked by a professional). In the attached Word document I listed some major content-related points for reflection, followed by minor points and line-by-line issues.

Author Response

Dear Ms. Natalija Knežević

Assistant Editor, MDPI Novi Sad

Reply to Reviewers’ Comments

Manuscript ID: sports-2072533

Article type: Original Research

Manuscript title: QUADRICEPS MUSCLE MORPHOLOGY IS AN IMPORTANT DETERMINANT OF MAXIMAL ISOMETRIC AND CRANK TORQUES OF CYCLISTS

Journal: Sports

Reply to reviewers’ comments

The authors would like to thank the Reviewers for the constructive comments, criticism, and feedback. All the reviewers’ comments have been addressed. In addition, edits to the revised manuscript have been added as tracked changes in the text.

attached the response letter to the reviewers

best regards

Reviewer 2 Report

This is an interesting article about “QUADRICEPS MUSCLE MORPHOLOGY IS AN IMPORTANT DETERMINANT OF MAXIMAL ISOMETRIC AND CRANK TORQUES OF CYCLISTS”.

This is a well designed study, and the conclusions are strongly supported by the data. Only a couple of suggestions:

Figure 2 is of poor quality.

I have not found the inclusion and exclusion criteria for the participants who participated in the study. If they had taken these criteria into account add it to the text.

Author Response

(The authors gave the same response as above.)

Round 2

Reviewer 1 Report

Dear authors,

I would like to congratulate the authors on the thorough revision of your manuscript and the extensive answers provided to my remarks. I hope this will result in an accepted paper, either as a great end-of-year accomplishment or a great start of the 2023.